# Numerical Study on Vibration Response of Compressor Stator Blade Considering Contact Friction of Holding Ring

**Jiaobin Ma [1], Zhufeng Liu [1], Di Zhang [2] and Yonghui Xie [1,\*]**

1   State Key Laboratory of Strength and Vibration of Mechanical Structures, School of Energy and Power Engineering, Xi'an Jiaotong University, Xi'an 710049, China; majiaobin@stu.xjtu.edu.cn (J.M.)
2   MOE Key Laboratory of Thermo-Fluid Science and Engineering, School of Energy and Power Engineering, Xi'an Jiaotong University, Xi'an 710049, China; zhang_di@xjtu.edu.cn
\*   Correspondence: yhxie@mail.xjtu.edu.cn

**Abstract:** There are many kinds of assembly structures in heavy-duty gas turbine compressor stator blades which have significant influence on the complex damping vibration characteristics. At present, compressor design is becoming more and more compact, so it is very meaningful to accurately obtain stator blade vibration characteristics of structures with contact damping. Firstly, a fretting slip friction dynamic model was introduced, and then a vibration analysis model of the compressor stator blade with outer ring structure was established based on the slip friction theory. Then, the vibration response of the compressor stator blade was obtained according to various working conditions, and the main factors affecting the vibration characteristics of the stator blade were revealed. Finally, the vibration response of the blade under a particular exciting force condition was simulated. The results show that the damping vibration characteristics of the compressor stator blades were affected by the excitation force and the normal load of the contact surface. The vibration response curve of the stator blade and the equivalent stiffness coefficient of the contact surface were analyzed, and the friction motion of the contact surface changed with the change of the working condition. The model can simulate the nonlinear vibration characteristics of stator blades. This method provides a reference for the vibration safety analysis of compressor stator blades.

**Keywords:** contact; compressor stator ring; friction; nonlinear vibration; exciting force





## 1. Introduction

In a compressor, the stator blade is a non-rotating component mounted on the cylinder. Generally, there are four types of assembly: cantilever stator blade assembly; assembled stator ring with inner and outer ring; welded stator inner and outer ring and variable mechanism stator blade. Due to the increase of friction contact between the stator blade and auxiliary structure, the study of vibration characteristics has become more complicated, and it is very necessary to study the friction damping problem with strong nonlinear characteristics. Under the current economic and environmental background, the design of heavy gas turbine compressors is becoming more and more compact, which greatly reduces the working clearance between parts [1–3]. Therefore, the contact interaction between stator blades and other structures also exists for a long time even under general working conditions.

Data show that accidents caused by the forced vibration of compressor blades account for more than 1/4 of the total number of blade accidents. Due to the large flow rate and long blade of the first-stage stator blade of heavy-duty gas turbine compressors, serious fatigue phenomenon and fracture failure are more likely to occur. The fracture accident of a compressor stator ring caused by blade high-cycle fatigue has been analyzed in detail in the literature [4]. Therefore, accurate analysis of airflow excitation characteristics and vibration response characteristics of compressor stator blades are of great significance for improving blade safety, reducing accident rates and improving the design level of compressors.

When compressor stator blades with attached structures are subjected to an excitation force, the adjacent working faces will slide relatively, and the resulting friction will consume vibration energy and thus affect vibration characteristics [5]. Therefore, the nonlinear characteristics of the contact surface will significantly affect the vibration response behavior of the stator blade, and tests show that this dry friction damping is related to a variety of factors [6].

The microslip model is widely used in the research of dry friction damping systems because it can simulate the actual friction contact behavior. Yang [7] proposed a two-dimensional dry friction contact model. Two massless springs were used to characterize tangential and normal contact stiffness, and two-dimensional frictional relative motion under variable normal load was studied. Chen and Menq [8] proposed a three-dimensional frictional contact model based on the two-dimensional model. Petrov [9,10] studied the vibration reduction characteristics of various types of dry-friction-damped structures by using a three-dimensional friction model. Based on Hertz theory, Burdekin et al. [11] verified the normal contact stiffness considering the local normal elastic deformation characteristics between elastic bodies through experiments, and established the tangential contact stiffness expression using Mindlin theory [12]. In this method, the influence of curvature radius and contact condition on contact stiffness was fully considered.

In general, the actual blade structure has a prominent feature; that is, the corresponding numerical finite element calculation model has more degrees of freedom. This also leads to the need for more computing power to accommodate such dynamic analysis, especially for high-complexity friction contact simulations. Many scholars have studied the nonlinear factors of rotating mechanical blades with contact friction characteristics. Cha [13] established an analytical method. The equivalent linearization method was used to deal with the nonlinear response and the correlation between vibration and friction damping of the model was studied by using statistical characteristics. Sinclair [14–16] concluded through research that the nonlinear characteristics of system vibration were caused by friction caused by contact structure deformation. Petrov [9] summarized effective methods for the study of blade disk structures with clearance and friction damping. The vibration response of the model was obtained by analyzing nonlinear periodic symmetries. Ender [17] analyzed the characteristics of blade system vibration under the centralized mass model and obtained the influence rule through the change of dry friction. Wang [18] constructed a damping model considering the dry friction effect and analyzed the vane disk response with the harmonic method, and obtained the variation rules of influencing factors such as coupling strength, viscous damping ratio, excitation force and friction amplitude.

Firrone [19] developed a method to solve the model forcing response. In this method, stator blade displacement and dynamic displacement are combined to calculate the displacement of blade disk system. Zhang [20] studied the vibration characteristics of a blade disk model with friction damping structure. The results show that the macroscopic slip caused by partial dry friction of the blade root can reduce the vibration amplitude. Ma [21] constructed a periodic sector structure to build a finite element model with frictional contact features. The influence of different factors on the characteristics of the contact area was studied, including centrifugal force, friction coefficient, installation angle and contact area. The results show that the friction coefficient had great influence on the contact pressure and displacement, but the installation angle had no obvious influence. Peng [22], Wang [23] and Chen [24] analyzed the stress distribution variation in the contact region under different structural clearance models. Zeng [25] analyzed the vibration characteristics of blade disc structure considering friction and clearance contact. The lumped parameter method was used to construct the nonlinear dynamic system to capture the obvious nonlinear dynamic behavior.

To the authors' knowledge, there is a great deal of research work on the vibration characteristics of different types of blade-damping structures, but there are few analyses on the inner and outer ring structures of compressor stator blades. At present, compressor design is becoming increasingly compact, so it is an important problem to accurately

obtain the vibration characteristics of stator blades considering the inner and outer ring damping structures. In terms of blade safety and compressor reliability design, this research direction needs further development. In this paper, the fretting slip friction dynamics model is introduced, and then the vibration analysis models of compressor stator blades with different inner and outer ring structures are established based on the slip friction theory. Further, the vibration response of compressor stator blades is obtained under various working conditions, and the main factors affecting the vibration characteristics of stator blades are revealed. The damped vibration rule of blades under inner and outer ring structures is summarized. A method for solving the nonlinear vibration of the compressor stator ring is provided.

## 2. Numerical Methods

### 2.1. Frictional Contact Model

Generally speaking, models for dry friction can be divided into two types. One is the macroscopic slip model, which is easy to understand and solve, but its scope of application is relatively narrow. Generally, macroscopic models can only be applied to dry friction structures with small normal load or large relative displacement. The other is the micro-slip model, which is characterized by the consideration of elastic deformation between contact surfaces, so it has a wider application and can simulate dry friction models more accurately.

At present, the contact state between blade-damping structures usually includes point to surface contact and surface to surface contact. In fact, the friction damping models applicable to different contact states are also very different, which cannot be characterized by a unified one. However, in the process of blade machining, there are some errors in machining accuracy. As a result, the blade-damping contact surface is usually characterized by point-to-surface contact. In this part, the Mindlin model, a fretting slip model that describes the point-to-surface contact state, is used to establish the contact relationship between blade-damping contact surfaces.

The micro-slip model is also called the local slip model. Different from the macroscopic slip model, it assumes that the stress and strain states of each node on the contact surface are not identical under the action of external forces. Under a large normal load, part of the contact area slides macroscopically and the rest is in a viscous local slip state. Compared with the macroscopic slip model, the micro-slip model can more accurately describe the friction-damping characteristics between contact surfaces.

Considering the elastic contact deformation of contact surfaces, the Mindlin microslip model is proposed to describe the frictional damping characteristics of a contact surface [26]. As shown in Figure 1, the relationship of friction with respect to displacement can be expressed as:

$$f(u) = \begin{cases} \mu N - 2\mu N \left\{ 1 - \left(\frac{1}{2}\right)^{\frac{3}{2}} \left[ 1 + \frac{2}{3A_0} \cdot \left( u - A + \frac{3}{2}A_0 \right) \right]^{\frac{3}{2}} \right\} & 0 \leq \theta \leq \theta^* \\ -\mu N & \theta^* \leq \theta < \pi \\ -\mu N + 2\mu N \left\{ 1 - \left(\frac{1}{2}\right)^{\frac{3}{2}} \left[ 1 - \frac{2}{3A_0} \cdot \left( u + A - \frac{3}{2}A_0 \right) \right]^{\frac{3}{2}} \right\} & \pi \leq \theta \leq \pi + \theta^* \\ \mu N & \pi + \theta^* \leq \theta < 2\pi \end{cases} \tag{1}$$

where $f$—friction force/N; $A$—displacement amplitude of the relative motion/μm; $A_0$—critical displacement amplitude of the relative motion/μm; $u$—displacement of the relative motion/μm; $\mu$—dynamic friction coefficient and $N$—normal forces of contact surfaces/N.

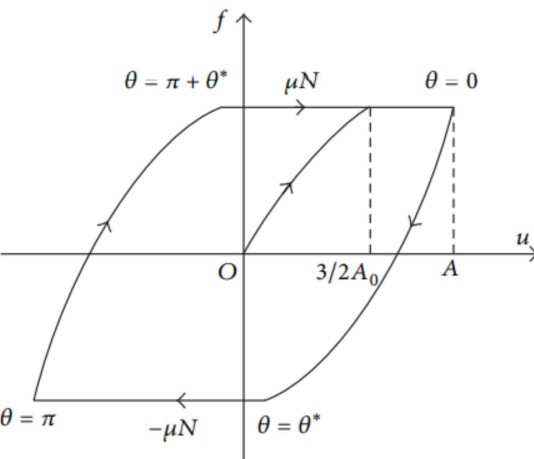

**Figure 1.** Mindlin microslip model.

It is assumed that the critical displacement $A_0$ is the ratio of the critical friction $\mu N$ and tangential stiffness $K_d$ on the contact surface, which can be described by the following formula.

$$A_0 = \frac{\mu N}{K_d} \tag{2}$$

When the material parameters of two touching objects are the same, the contact radius $a$ is described by the following formula.

$$a = \left[\frac{4}{3}\pi \times \frac{N \times (k_1 + k_2) \times R_1 \times R_2}{R_1 + R_2}\right]^{1/3} \tag{3}$$

$$k_1 = \left(1 - v_1^2\right)/(\pi E_1) \tag{4}$$

$$k_2 = \left(1 - v_2^2\right)/(\pi E_2) \tag{5}$$

where $a$ is the radius of the contact area/m; $N$ is the positive contact pressure over $N$; $R_1$ is the radius of the first contact pair/m; $R_2$ is the radius of the second contact pair over m; $v_1$ is the first contact to Poisson ratio; $v_2$ is the second contact to Poisson ratio; $E_1$ is the elastic modulus of the first contact pair/Pa and $E_2$ is the elastic modulus of the second contact pair/Pa.

The tangential contact stiffness $K_d$ of the contact pair can be calculated by the following formula.

$$K_d \approx K_\infty = \frac{4Ga}{2 - v} \tag{6}$$

where $G$ is shear modulus/N·m$^{-2}$ and $v$ is the Poisson ratio of the object.

Based on Hertz theory, the shear modulus of this type of contact region is defined as

$$G = \frac{E}{2(1 + v)} \tag{7}$$

### 2.2. Vibration Response Analysis Method

Based on the Mindlin model, the dynamic response analysis of the friction damping contact surface of the compressor stator blade is realized. The contact surface between the blade and the outer ring is connected by a spring damping element to construct the finite element model [27]. It should be noted here that the equivalent stiffness coefficient $K_{eq}$ and equivalent damping coefficient $C_{eq}$ are defined on the model contact surface according to the geometric position of the element. By averaging the equivalent stiffness and damping

coefficients under a certain normal load N, tangential stiffness and damping coefficients of each element matrix are obtained as shown in Figure 2.

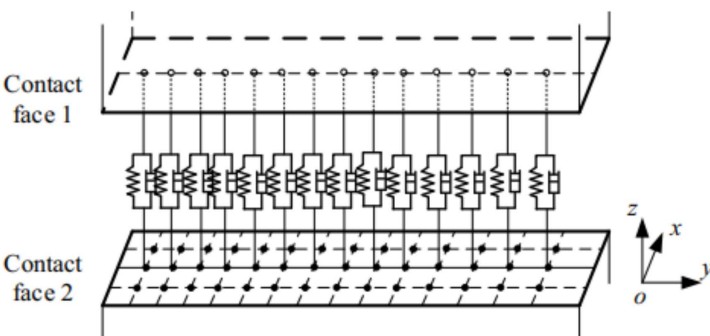

**Figure 2.** Spring damping element.

The elastic and kinematic responses of the spring-damping element are determined by the stiffness coefficient of the stiffness matrix element and the damping coefficient of the damping matrix element. The stiffness matrix **K** and damping matrix **C** of the spring damping element are $12 \times 12$ matrices. Suppose $x$, $y$ and $z$ are the directions on the node, then the stiffness matrix **K** and damping matrix **C** are:

$$\mathbf{K} = \begin{bmatrix} k_x & 0 & 0 & 0 & 0 & 0 & -k_x & 0 & 0 & 0 & 0 & 0 \\ 0 & k_{\tau y} & 0 & 0 & 0 & 0 & 0 & -k_{\tau y} & 0 & 0 & 0 & 0 \\ 0 & 0 & k_{\tau z} & 0 & 0 & 0 & 0 & 0 & -k_{\tau z} & 0 & 0 & 0 \\ \vdots & \vdots & \vdots & \vdots & \vdots & \vdots & \vdots & \vdots & \vdots & \vdots & \vdots & \vdots \\ -k_x & 0 & 0 & 0 & 0 & 0 & k_x & 0 & 0 & 0 & 0 & 0 \\ 0 & -k_{\tau y} & 0 & 0 & 0 & 0 & 0 & k_{\tau y} & 0 & 0 & 0 & 0 \\ 0 & 0 & -k_{\tau z} & 0 & 0 & 0 & 0 & 0 & k_{\tau z} & 0 & 0 & 0 \\ \vdots & \vdots & \vdots & \vdots & \vdots & \vdots & \vdots & \vdots & \vdots & \vdots & \vdots & \vdots \\ 0 & 0 & 0 & 0 & 0 & 0 & 0 & 0 & 0 & 0 & 0 & 0 \end{bmatrix} \tag{8}$$

$$\mathbf{C} = \begin{bmatrix} c_x & 0 & 0 & 0 & 0 & 0 & -c_x & 0 & 0 & 0 & 0 & 0 \\ 0 & c_{\tau y} & 0 & 0 & 0 & 0 & 0 & -c_{\tau y} & 0 & 0 & 0 & 0 \\ 0 & 0 & c_{\tau z} & 0 & 0 & 0 & 0 & 0 & -c_{\tau z} & 0 & 0 & 0 \\ \vdots & \vdots & \vdots & \vdots & \vdots & \vdots & \vdots & \vdots & \vdots & \vdots & \vdots & \vdots \\ -c_x & 0 & 0 & 0 & 0 & 0 & c_x & 0 & 0 & 0 & 0 & 0 \\ 0 & -c_{\tau y} & 0 & 0 & 0 & 0 & 0 & c_{\tau y} & 0 & 0 & 0 & 0 \\ 0 & 0 & -c_{\tau z} & 0 & 0 & 0 & 0 & 0 & c_{\tau z} & 0 & 0 & 0 \\ \vdots & \vdots & \vdots & \vdots & \vdots & \vdots & \vdots & \vdots & \vdots & \vdots & \vdots & \vdots \\ 0 & 0 & 0 & 0 & 0 & 0 & 0 & 0 & 0 & 0 & 0 & 0 \end{bmatrix} \tag{9}$$

$$k_x = \frac{K_n}{n}; \ k_{\tau y} = \frac{K_{eq}}{n}; \ k_{\tau z} = \frac{K_{eq}}{n} \tag{10}$$

$$c_x = 0; \ c_{\tau y} = \frac{C_{eq}}{n}; \ c_{\tau z} = \frac{C_{eq}}{n} \tag{11}$$

where $K_n$ is the normal contact stiffness between contact surfaces/N·m$^{-1}$ and $n$ is the number of spring-damping units between contact surfaces.

For the friction-damping model studied in this paper, the response obtained by the numerical solution of the harmonic method is also harmonic when subjected to harmonic excitation. In order to reduce the calculation workload, only the first harmonic is considered in the friction force. Assuming that the relative displacement is $A\cos\theta$, the friction interface

in the contact region is simplified into a spring damping model. Take the elastic force and damping together to obtain the friction force.

$$f = f_c(A) \cos \theta + f_s(A) \sin \theta = K_{eq} A \cos \theta - C_{eq} \omega A \sin \theta \tag{12}$$

The characteristic parameters of system vibration include displacement, frequency and phase. In the formula, they are represented by $A$, $\omega$ and $\theta$, respectively. The equivalent coefficient of contact surface can be obtained by the following formula.

$$K_{eq} = \frac{f_c(A)}{A} = \frac{1}{\pi A} \int_0^{2\pi} f(A, \theta) \cos \theta d\theta \tag{13}$$

$$C_{eq} = -\frac{f_s(A)}{\omega A} = -\frac{1}{\omega \pi A} \int_0^{2\pi} f(A, \theta) \sin \theta d\theta \tag{14}$$

The calculation method of the blade vibration response is given in Figure 3. It should be noted that the calculation in this study requires a large number of iterative processes to obtain the final result. A brief summary includes the following steps: (1) First, the initial normal load should be defined for the contact surface. (2) Frequency is defined. (3) Then, assume the initial displacement of a region. (4) The equivalent coefficient is obtained by using the above model and formula. (5) According to the damping and stiffness coefficients, the finite element method is used to solve the displacement of the model. (6) At this time, it is necessary to judge whether the solution process meets the convergence standard. If the iterations do not converge, then respecify the initial displacement and return to step 3. If the iterative process converges, then update the frequency and return to step 2. (7) Further, the different normal loads can be set for solving, and then return to step 1 to restart the iterative process. (8) After solving the required normal load conditions, the numerical analysis is completed.

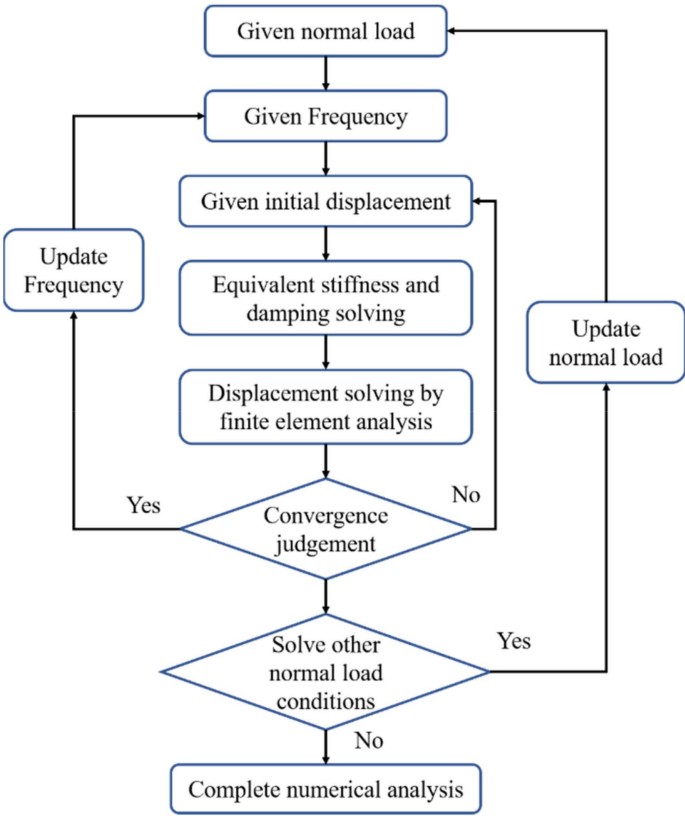

**Figure 3.** Iterative process of calculation.

### 2.3. Airflow Load Analysis

In this paper, the numerical simulation of the compressor model flow adopted a three-dimensional unsteady calculation method. In the long-term industrial practice of compressor model flow analysis, it is an effective means to obtain the internal flow field parameters by means of the Reynolds average method. The basis of numerical simulation is the fluid dynamics equation, including mass, momentum and energy conservation as the three main equations. The mass equation is as follows:

$$\frac{\partial \rho}{\partial t} + \frac{\partial(\rho u)}{\partial x} + \frac{\partial(\rho v)}{\partial y} + \frac{\partial(\rho w)}{\partial z} = 0 \tag{15}$$

The momentum equation is as follows:

$$\frac{\partial U}{\partial t} + \frac{\partial(E - E_v)}{\partial x} + \frac{\partial(F - F_v)}{\partial y} + \frac{\partial(G - G_v)}{\partial z} = 0 \tag{16}$$

The energy equation is as follows:

$$\frac{\partial(\rho T)}{\partial t} + div(\rho U T) = div\left(\frac{\lambda}{C_p} \cdot gradT\right) + S_T \tag{17}$$

The *k-w* turbulence model was used in this study and ideal gas was used as the working medium. The gas equation of state is defined by the following formula.:

$$p = \rho R T \tag{18}$$

where $\rho$—density/kg·m$^{-3}$; $U$, $u$, $v$, $w$—velocity and its coordinate components/m·s$^{-1}$; $t$—time/s; $T$—temperature/K; $E$, $F$, $G$—convectional momentum flux/kg·s$^{-2}$·m$^{-1}$; $E_v$, $F_v$, $G_v$—viscosity momentum flux/kg·s$^{-2}$·m$^{-1}$; $C_p$—specific heat/J·kg$^{-1}$·K$^{-1}$ and $\lambda$—heat transfer coefficient/ W·m$^{-2}$·K$^{-1}$.

## 3. Computational Model

### 3.1. Finite Element Model of Compressor Stator Blade

Compressor stator blades are generally installed in combination with structures such as holding rings and sealing rings during operation, and the contact position may be tightened by welding or bolted connection. In this paper, the finite element method was used to model the blade to achieve the three-dimensional nonlinear vibration simulation. It is difficult to solve the nonlinear vibration of the contact surface. In order to shorten computation time and save computation resources, nonlinear vibration research is concentrated in a blade sector.

In this paper, a whole circle model of a compressor stator blade with an external holding ring was established, and then a sector was intercepted as the research object for nonlinear vibration analysis. Unsteady aerodynamic numerical studies of this model were carried out in [28], and vibration analysis was further carried out in this paper. As there are some structures, such as stop pins in the stator blade ring, which restrict some circumferential movement in general, the calculation results can be obtained using sector division. Figure 4 shows the whole circle of the stator blade ring model, and Figure 5 shows a specific stator blade sector model. The stator blade contacts the outer holding ring through the tip barb, which is fixed on the cylinder and other structures, and the pre-tightening force is exerted on the contact surface through fastening bolts. The whole structure is excited by the aerodynamic load of the blade surface, with the nonlinear characteristics due to the friction contact between the components. The boundary conditions of the model are described below. The red and blue lines represent the surface at that position. The displacement degree of freedom constraint was defined for the top surface nodes. The two contact surface nodes of the stator blade define the spring damping element. The

surface center nodes represented by the blue line define the degree of freedom coupling. In addition, blade surface nodes apply airflow force boundaries.

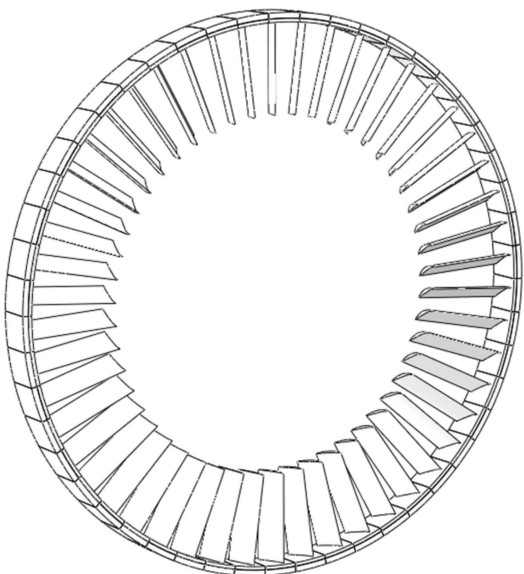

**Figure 4.** Full circle compressor stator blade ring model.

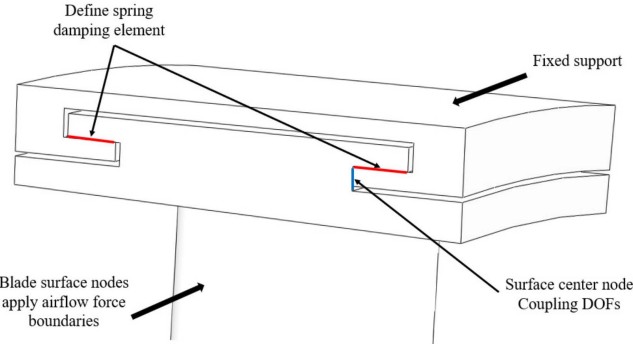

**Figure 5.** Stator blade sector contact friction model.

The number of stationary blades in the circle was 50, and the height of the blades was 230 mm. One side of the blade was in contact with the outer ring, and the other side was the free end. The blade diameter was 565 mm. The axial length of the blade was 73 mm. Young's modulus was 207 GPa, Poisson's ratio was 0.3 and density was 7850 kg/m$^3$. Some of the parameters used here are commonly used parameter ranges in engineering. Aerodynamic excitation force was applied to the blade surface. In ANSYS, solid185 elements were used in combination to simplify the overall structure into a finite element model of stator blade nonlinear vibration, in which a few solid186 elements were used in some excessive areas.

Due to the nonlinear action, the main area of action was the contact surface between the outer ring and the tip barb of the stationary blade. In this area, a series of contact surface acting elements were established to simulate the influence of friction damping by using spring damping elements, and the model was carried out as a whole. The finite element model is shown in Figure 6. The analysis model was composed of 5062 nodes, and 18 spring damping elements were defined between the contact surface. When the update error was less than 0.5%, the calculation results were deemed as convergence. Finally, the vibration response curves of the stator blade were obtained.

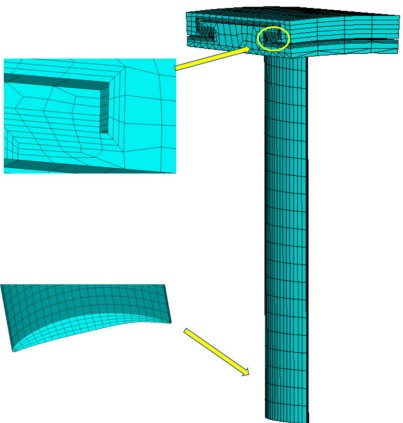

**Figure 6.** Stator blade finite element model.

*3.2. Stator Blade Load Analysis Model*

In this paper, a numerical model of unsteady flow field was constructed to obtain the flow load on stator blades. Due to the high complexity of the transient solution, in order to minimize the computing resources and time under the condition of enough accuracy, multi-core calculation was used to improve efficiency and the number of blades in the model was fine-tuned. The number of guide blades, rotor blades and stator blades were 50/25/50, respectively. The transient analysis model of 1.5 stage compressor flow field established in this way can quickly and accurately obtain the flow load on the compressor stator blade surface. The rotating speed of the rotor blade was 3000 rpm. The unsteady flow excitation data of compressor stator blades can be obtained under the given working conditions. The research model is shown in Figure 7.

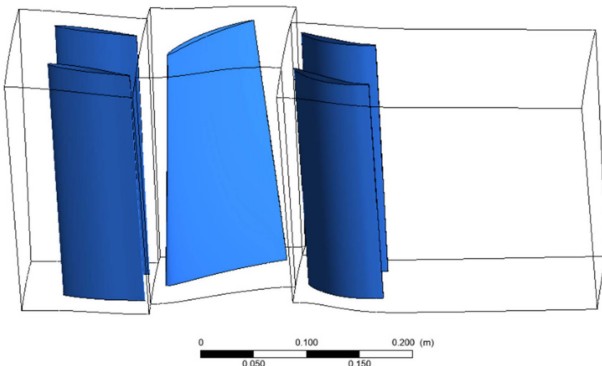

**Figure 7.** The transient flow numerical model.

The grid division of the numerical model was completed using TurboGrid. It should be noted that axial extension was carried out appropriately for the downstream of the stator blade model to improve the convergence of the numerical simulation. The O-mesh method was mainly used for meshing, which showed good boundary layer quality improvement for the area near the blade surface. At the same time, in order to effectively improve the accuracy of the solution, the density of the local grid was improved, especially at the leading edge and trailing edge of the blade. The number of grids in the whole transient flow calculation model was $4.4 \times 10^5$. Figure 8 shows a grid diagram of the fluid domain during CFD calculation. The boundary conditions were calculated using total temperature and pressure at the entrance and mass flow at the exit. Corresponding to a working condition point of the compressor model, the total inlet pressure was 75,000 Pa. The total temperature of import was 300 K. The outlet mass flow rate was 72 kg/s.

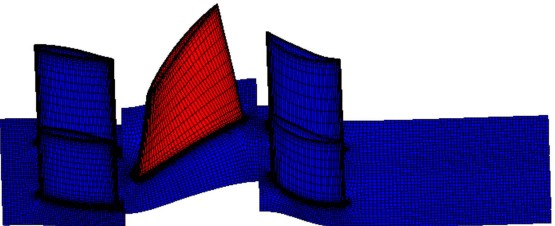

**Figure 8.** Mesh of rotor and stationary blades.

## 4. Results and Discussion

### 4.1. Nonlinear Vibration Results of Different Models

The nonlinear vibration of compressor stator under the action of normal load and exciting force was simulated with iterative calculation. For the analysis of the model, the vibration response characteristics of the blades were analyzed by changing the normal load and assuming that the exciting force remained unchanged. The circumferential flow force and axial flow force of the blade surface were set as 40 N and 16 N, respectively. The vibration amplitude of the outlet node of the 50% blade height was extracted. Figure 9 shows the vibration response curves with a series of different normal loads under corresponding working conditions.

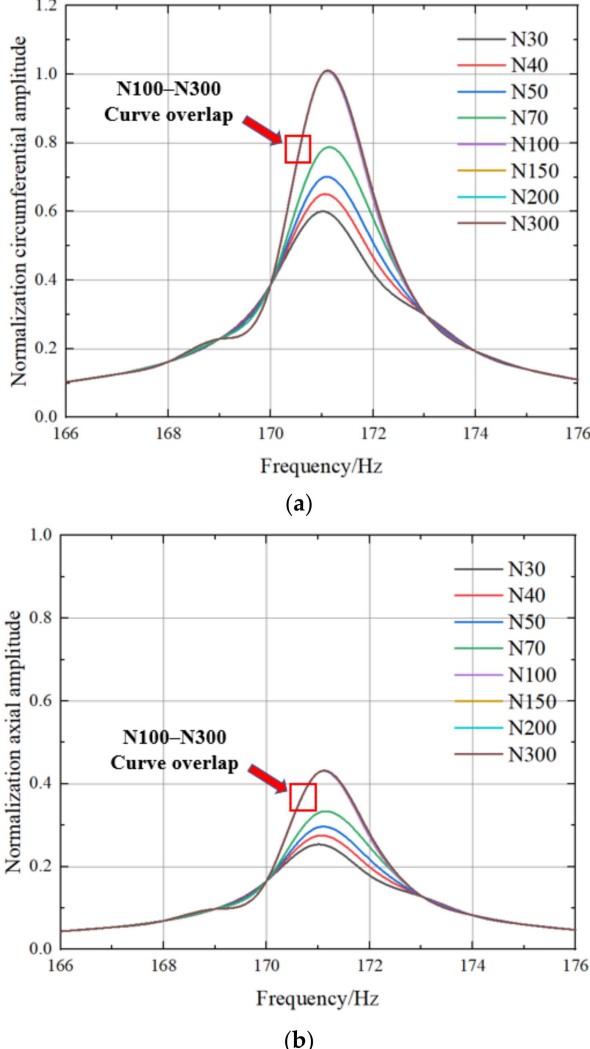

**Figure 9.** Response normalization amplitude curves under different normal loads with 40 N exciting force. (**a**) Circumferential amplitude curve. (**b**) Axial amplitude curve.

Since the blades were subjected to airflow forces in two directions, the results given here include circumferential and axial vibration responses. It is worth pointing out that the vibration response curves of several results with high normal load in Figure 9 are overlapped, which made it impossible to distinguish in detail. The reasons will be analyzed in detail later. In order to facilitate comparison, the amplitude was normalized. The maximum amplitudes calculated from the resonant frequencies in a series of working conditions were taken as the reference to normalize the axial and circumferential curves.

Firstly, it is not difficult to see that when excitation force remains constant, the vibration response of the compressor stator blade increases with the increase of normal load, and the vibration response no longer changes when the normal load increases to a certain extent. This phenomenon can be observed for both axial and circumferential amplitude. In addition, the resonant frequencies of various calculation conditions are roughly in the same position. At the same time, the ratio of blade amplitude magnitude of circumferential and axial is roughly consistent with the ratio of excitation force given in the corresponding direction.

As for the phenomenon of blade vibration response no longer changing significantly under the normal load of a large contact surface, we can provide the following explanations. Under dry friction damping, the normal load of blade contact surface changes the motion state of the blade contact surface. When the normal load is small, the friction motion of the contact surface between the blade and the outer ring is in a macroscopic slip state. In this case, the friction damping effectively dissipates the vibration energy, thus reducing the common amplitude value of the blade. With the increase of normal load, tangential stiffness of contact surface increases, friction motion converts into fretting slip state and the vibration energy dissipated by friction damping is very small. When the normal load reaches a certain level, the contact surface can be approximated as no longer having relative motion and the blade and outer ring are approximately seen as a whole. At this time, the change of normal load no longer affects the amplitude of blade vibration response.

The stator blade vibration response curve obtained in this paper can be compared with that in [27]. In this reference, the vibration response of the blade disk was analyzed using the Mindlin contact friction model. Similar rules appear in the process of numerical analysis. When the normal load and excitation force change, the motion state of the contact surface also changes between macroscopic slip and micro slip. This shows that the research of this paper is reasonable.

In order to further analyze the changing characteristics of the contact surface, in the iterative calculation, the variation of the equivalent stiffness coefficient of the contact surface with the normal load of the contact surface under resonant frequency was extracted. Here, because the range of normal load and equivalent stiffness coefficient varied by a large order of magnitude, logarithmic coordinates were adopted for the convenience of analysis.

The following Figure 10 shows the variation trend of the equivalent stiffness coefficient with normal load under the 40 N excitation force model. It can be clearly seen from the figure that under a certain normal load, the equivalent stiffness coefficient of the contact surface has a step feature. When the normal load is small, the equivalent stiffness coefficient increases slowly. When the normal load reaches a certain value, the equivalent stiffness coefficient hardly changes; that is to say, under certain excitation force conditions, the small normal load contact surface is a macro slip state. When the normal load increases to a certain value, the contact surface state changes to fretting slip or even almost no slip, because at this time, very large normal load leads to huge tangential stiffness. This phenomenon corresponds with the previous analysis.

In fact, the compressor stator blade will be affected by a variety of airflow excitation forces of different orders during operation. Generally speaking, the amplitude of first-order airflow force is larger, while that of higher-order airflow force is smaller, but its effect can also be analyzed to some extent. Here, the airflow excitation force level of the model was changed, and the circumferential airflow force and axial force of the blade surface were set as 4 N and 1.6 N. We did not change the ratio of the flow force in the two directions for the sake of calculation rationality. Similarly, we give the vibration response curves

under a series of different normal contact loads under corresponding working conditions in Figure 11 and the amplitude curve is normalized here.

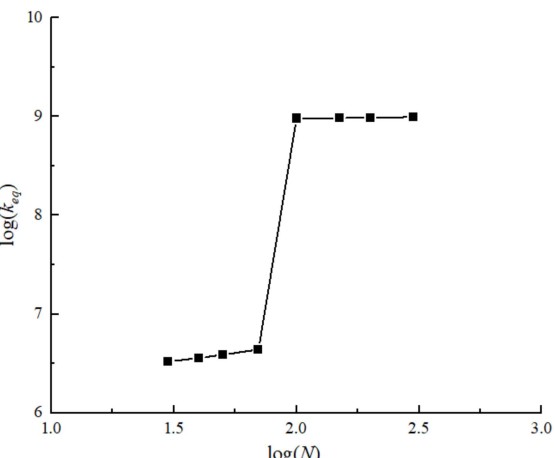

**Figure 10.** The equivalent stiffness coefficients of different normal loads at the resonant frequency of 40 N excitation force.

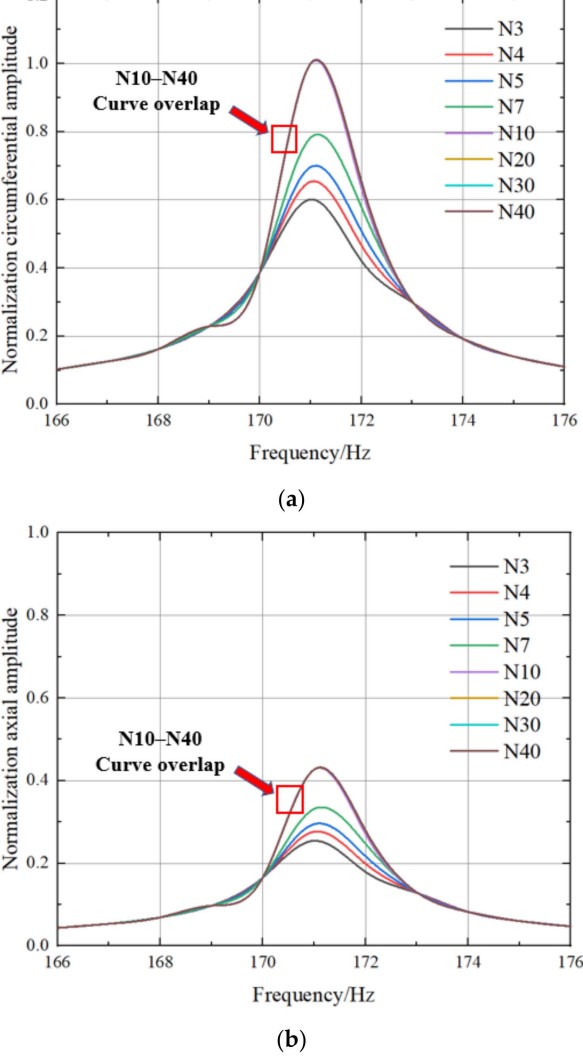

**Figure 11.** Response normalization amplitude curves under different normal loads with 4 N exciting force. (**a**) Circumferential amplitude curve. (**b**) Axial amplitude curve.

Of course, the normal load of the contact surface set by a small airflow force amplitude cannot be too large, otherwise the contact surface is in an approximate stick state. From the analysis figure, we can first find a similar trend with the 40 N excitation force level; that is, the vibration response increases with the increase of normal load. However, it still increases to a point where it does not change. The reason for this is similar to the previous analysis, except that because the excitation force is small, the normal load required for the state transition of the contact surface is also very small. In addition, the resonance frequency is in the same position as previously calculated.

Again, the equivalent contact stiffness curve is given here in Figure 12. It can be seen that the contact surface of the model still has a frictional state change under the lower excitation force level. However, the excitation force corresponding to the normal load is different. Logarithmic coordinates were used again because the range of equivalent stiffness coefficients was too large. The analysis of this part is similar to the previous analysis, and will not be repeated in the paper.

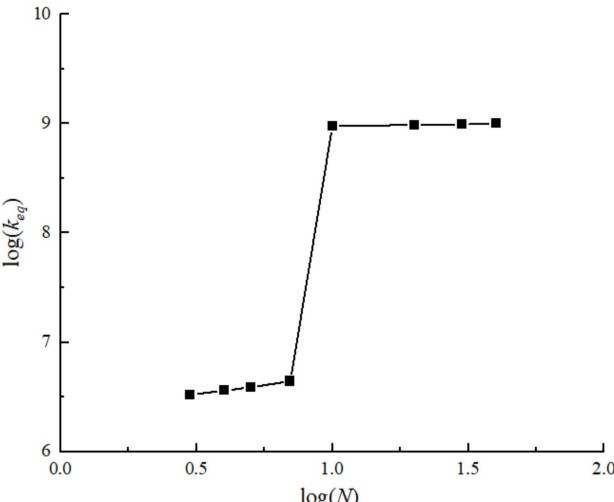

**Figure 12.** The equivalent stiffness coefficients of different normal loads at the resonant frequency of 4 N excitation force.

Previously, the nonlinear vibration response characteristics of the model contact surface under different excitation force levels were analyzed. Another case is studied here; that is, the normal contact surface load was constant, and then the amplitude of the excited force was changed. The process analyzed is similar to the previous one. Using the same numerical model, the normal load of the contact surface was assumed to be 30 N. Then, the displacement amplitude of the 50% high outlet side of the stator blade was extracted by changing the excitation force of a series of blades. The ratio between the axial and circumferential flow forces of the model and the excitation force was still maintained during the analysis.

The following Figure 13 shows the curve of the blade vibration response. It can be seen that under the same normal load on the contact surface, the amplitude of the blade vibration response increases with the increase of airflow excitation force. This trend is easy to understand, because a larger airflow excitation force contains more vibration energy, which corresponds to a larger vibration amplitude. However, the contact characteristic of the contact surface is a nonlinear characteristic, which can be identified by careful analysis of the figure here. When the excitation force is small, the amplitude changes of the axial and circumferential vibration response and the magnitude changes of the excitation force basically show linear growth. However, when the exciting force increases to a certain order of magnitude, it is not difficult to see that the increasing magnitude of the exciting force is no longer linearly correlated with the increasing magnitude of the vibration amplitude. This indicates that the contact surface has nonlinear characteristics. It is reminiscent of the

previous analysis showing that the contact surface changes from fretting slip to macroscopic slip when the exciting force increases to a certain level. Some of the vibration energy is consumed by friction, so the amplitude change is no longer linear. It also shows that the nonlinear vibration characteristics of stator blades are simulated by this method.

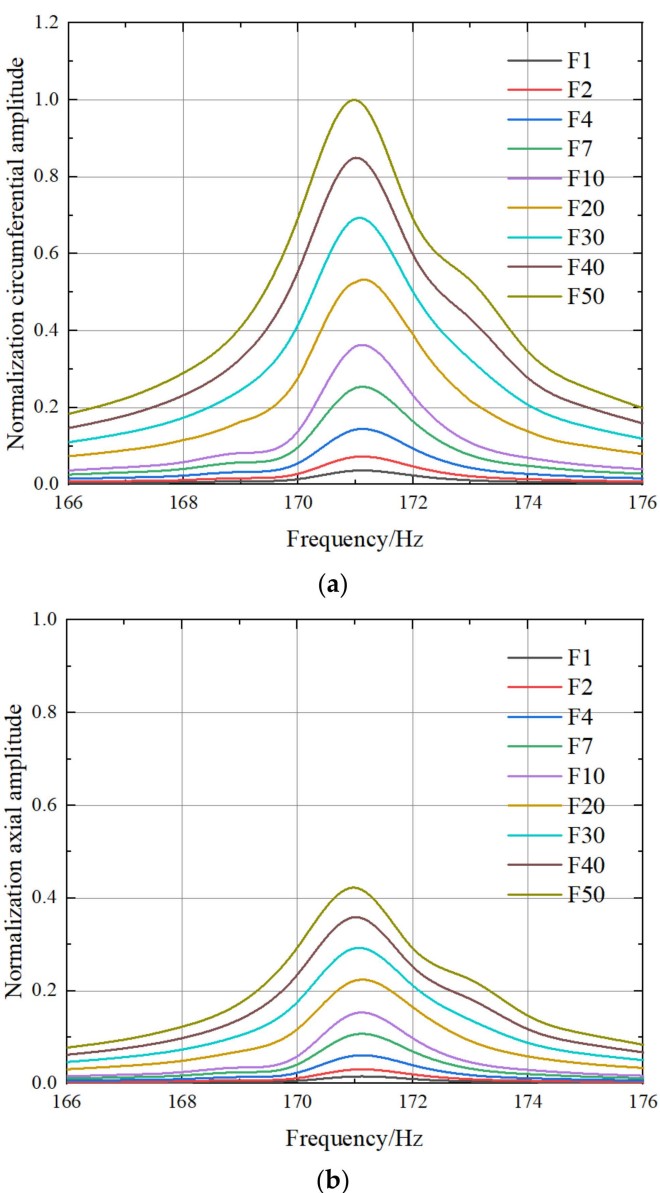

**Figure 13.** Response normalization amplitude curves under different exciting force with 30 N normal load. (**a**) Circumferential amplitude curve. (**b**) Axial amplitude curve.

The vibration responses of stator blades under various normal loads and excitation forces are solved. A very important phenomenon needs to be briefly discussed here. The sensitivity of the model studied in this paper to input parameters is critical. It can be seen from the above analysis that when the airflow force load is defined as a constant, the stator blade response curve is less sensitive to the larger normal load of the contact surface. When the normal load is in the lower parameter range, the response curve has strong sensitivity to the change of normal load. Of course, the normal load cannot be defined as too small, otherwise the computational model will not converge. Similarly, if the normal load is defined as a constant, then the change in the flow force directly affects the change in the response curve. In other words, the vibration response of the stator blade is very sensitive to the airflow force parameters.

In order to visualize this change, the variation curve of the equivalent stiffness coefficient is given in Figure 14. The logarithmic coordinates are also applied in the figure. The curve change here is just the opposite of the tendency of the constant excitation force to change the normal load. When the normal load of the contact surface is fixed at a certain level, the friction motion state of the contact surface also changes with the increase of the airflow excitation force to a certain value. It changes from viscous or fretting slip to macroscopic slip; therefore, the equivalent stiffness coefficient also takes a step. This phenomenon gives an important hint that the level of exciting force in this operation should be particularly considered during the installation of stator blades, because once the stator ring is assembled and running, its contact surface is kept tight by the normal pressure provided by the fixing bolt and other structures. However, if a wide range of conditions, such as variable operating conditions, occur during operation, blade vibration under some exciting forces under design conditions cannot be considered alone. Because of the high parameters of some operating conditions, the larger vibration force is likely to cause a blade contact surface friction state change.

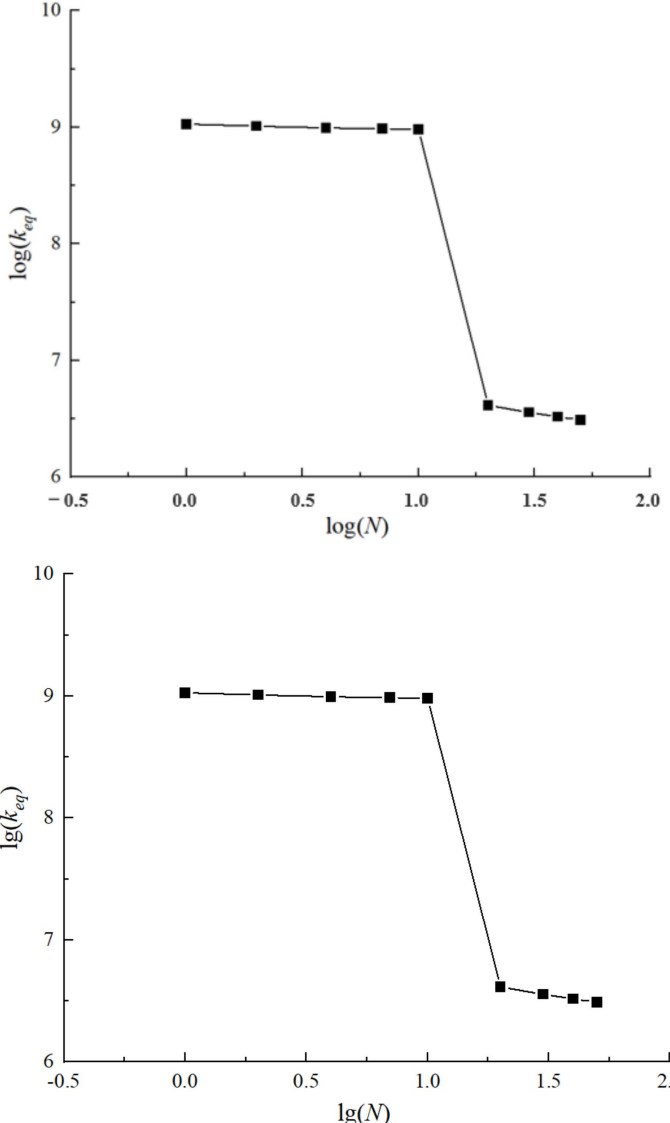

**Figure 14.** The equivalent stiffness coefficients of different exciting force at the resonant frequency of 30 N normal load.

### 4.2. Vibration Response under a Particular Exciting Force

The theoretical analysis of the compressor stator blade with an outer ring structure is discussed above. The influence of normal contact load and air excitation force on nonlinear vibration was studied. In this paper, the fluid–structure coupling analysis method was used to give the vibration response calculation results of compressor stator blades under a specific operating condition.

Firstly, the transient airflow excitation force analysis was carried out under a compressor design condition to obtain the unsteady airflow load on the stator blade surface. The following Figure 15 shows the time domain diagram of the flow force on the blade, including the axial and circumferential flow force. Furthermore, FFT is applied to the airflow force of the blade. The airflow force frequency domain results of the blade are shown in Figure 16. The airflow force corresponding to each order can be obtained. It can be seen that periodic airflow loads on the surface of the stator blades occur due to the wake action of the upstream blades during the normal operation of the compressor. After spectrum analysis, the relationship between the first-order frequency 1250 Hz and the number of rotor blades accords with the theory.

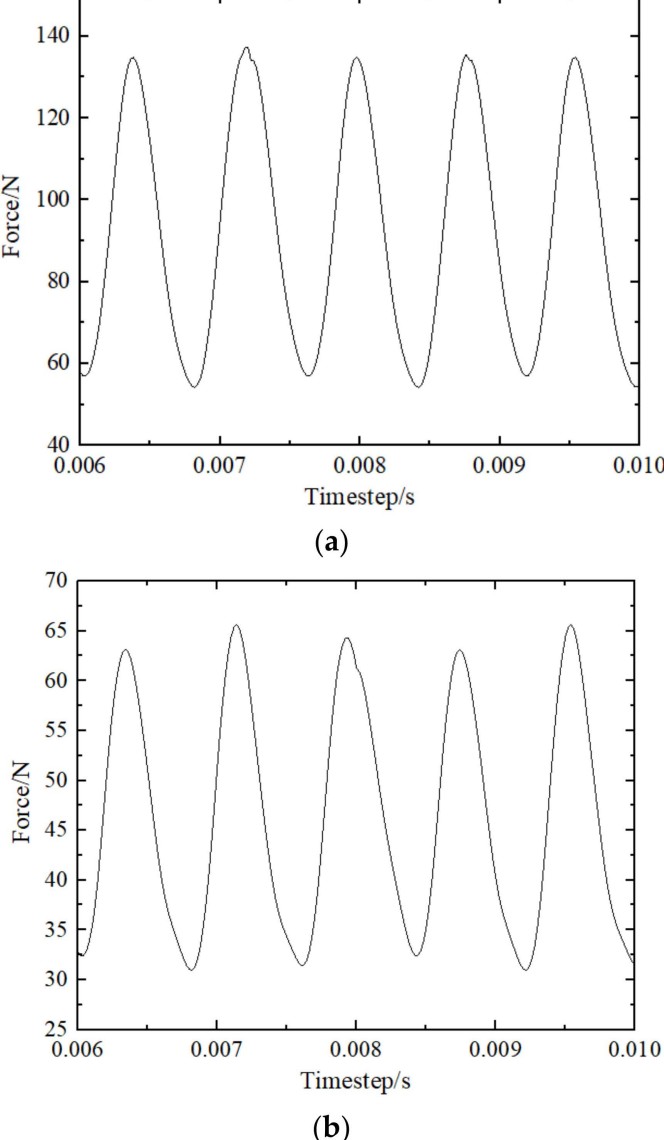

**Figure 15.** Compressor stator blade airflow excitation force. (**a**) Circumferential force. (**b**) Axial force.

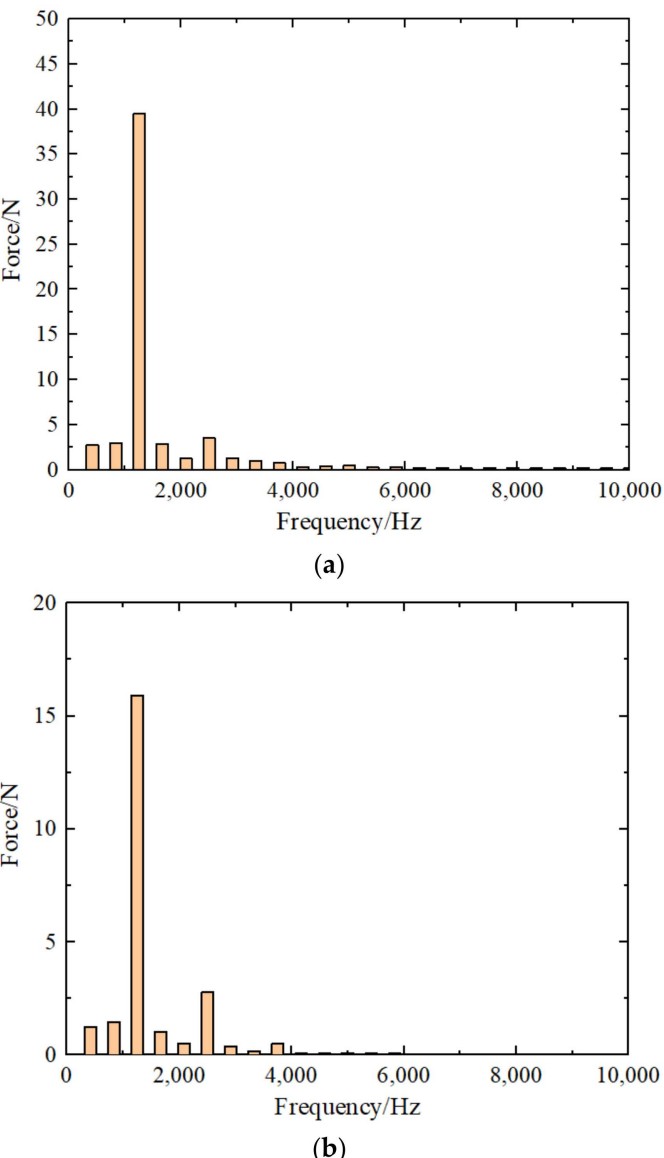

**Figure 16.** Exciting force FFT results. (**a**) Circumferential force. (**b**) Axial force.

The first-order flow force was used as the boundary of finite element calculation to simulate the vibration response characteristics of compressor stator blades under these working conditions. The normal load of the contact surface was set at 30 N. The vibration response of the 50% high of the stationary blade at the outlet side position is given here. The following Figure 17 shows the axial and circumferential vibration response curves of the air outlet node at the free end of the blade. It can be seen that the maximum circumferential vibration response value is 1.45 mm and the maximum axial response amplitude is 0.61 mm. Therefore, it is necessary to pay attention to the vibration safety of stator blades during the operation of equipment.

In general, there may be a certain range of changes in the working conditions of a compressor during its operation. The change of working condition is closely related to the airflow force on the compressor stator blade. This means that the aerodynamic load of the blade varies in a certain range during operation. However, the normal load of the blade contact surface is roughly at a constant level, which is determined by the mechanical structure of the equipment. According to the research method used in this paper, a better normal load of stator blade contact surface can be identified for design and installation. In brief, we aim to find a normal load which can not only meet the requirements of vibration

safety but also reduce the amplitude by analyzing the working conditions of a variety of different input parameters.

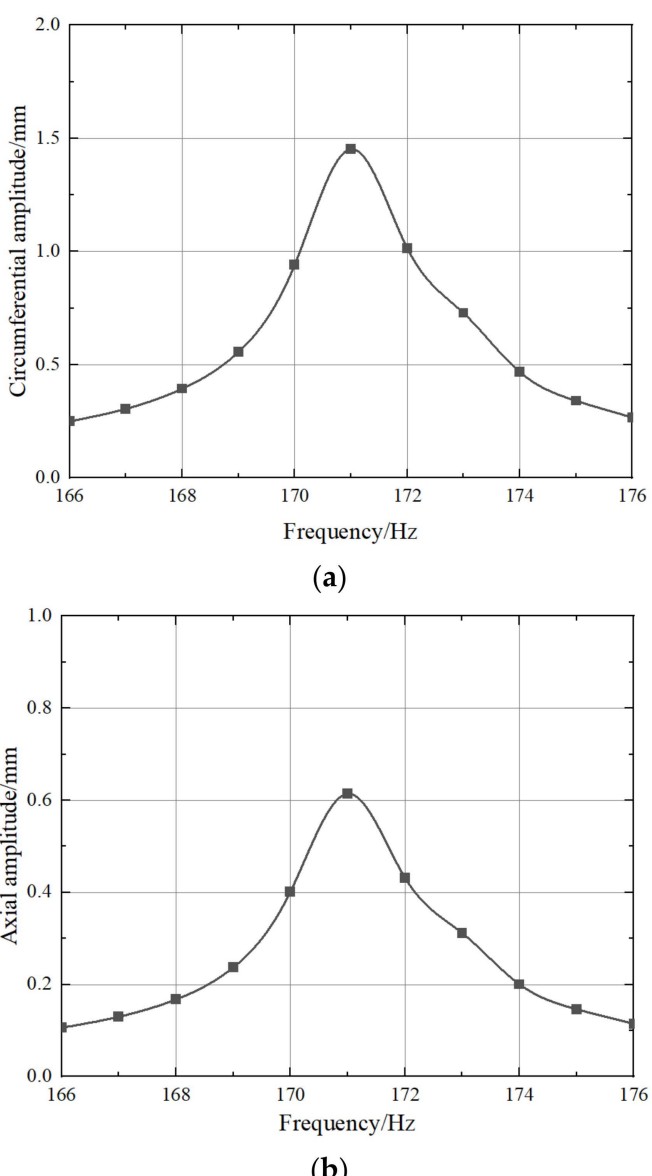

**Figure 17.** Vibration response curve of stator blade under this condition. (**a**) Circumferential amplitude curve. (**b**) Axial amplitude curve.

## 5. Conclusions

The vibration characteristics of the stator blades of a compressor with an external holding ring considering contact friction were studied in this paper. Firstly, the fretting slip friction dynamic model was introduced, and then the vibration analysis models of compressor stator blades with different inner and outer ring structures were established based on the slip friction theory. Then, the vibration response of compressor stator blades was solved according to various working conditions, and the main factors affecting the vibration characteristics of stator blades were revealed. Finally, the vibration response characteristics of compressor stator blades under a specific working condition were simulated. The damped vibration rule of blade under inner and outer ring structures were summarized.

The vibration responses under different normal loads of 40 N and 4 N and the vibration responses under different normal loads of 30 N were studied. When the blade excitation force is constant, the amplitude of vibration increases with the increase of normal load in a

certain range. When the normal load reaches a certain value, the blade vibration amplitude remains unchanged. When the normal load of blade is constant, the amplitude of vibration increases with the increase of excitation force in a certain range. When the exciting force is small, the proportion of the increase in amplitude is basically linear. However, it becomes nonlinear when the excitation force is large. Under the action of friction damping, the contact surface is affected by both normal load and exciting force. The friction state of the blade contact surface may convert between fretting slip and macroscopic slip condition. Thus, the nonlinear behavior of blade vibration response amplitude is obtained. The variation of the equivalent stiffness coefficient of the contact surface with the parameters also reflects this characteristic. In addition, this method is used to simulate the vibration response of stator blades under different conditions under certain assumptions. The model provides a reference for the study of vibration safety of compressor stator blades.

**Author Contributions:** Conceptualization, J.M.; Methodology, Z.L. and Y.X.; Validation, Z.L. and Y.X.; Formal analysis, J.M.; Investigation, J.M.; Resources, D.Z.; Writing—original draft, J.M.; Writing—review & editing, Z.L. and Y.X.; Visualization, J.M. All authors have read and agreed to the published version of the manuscript.

**Funding:** This research was funded by National Science and Technology Major Project grant number [J2019-IV-0022-0090].

**Acknowledgments:** All authors in the paper express great gratitude for the financial support by National Science and Technology Major Project (J2019-IV-0022-0090).

**Conflicts of Interest:** The authors declare that they have no known competing financial interests or personal relationships that could have appeared to influence the work reported in this paper.

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
