# Peer review of "Numerical Study on Vibration Response of Compressor Stator Blade Considering Contact Friction of Holding Ring"

_applsci, doi:10.3390/app13116380_

Round 1

Reviewer 1 Report

The author has conducted a numerical study on the vibration response of compressor stator blades, taking into consideration the contact friction of the holding ring. The paper provides a comprehensive analysis of the vibration characteristics of compressor stator blades with different inner and outer ring structures, based on the slip friction theory. The study highlights the main factors that affect the vibration response of the stator blades and provides valuable insights into the nonlinear behavior of blade vibration response amplitude.

However, I have a few comments

·       Provide the references for bladed size and blade material selection. This would give readers a better understanding of the input parameters and how they were obtained.

·       Provide more details on the computational setup and boundary conditions: It would be helpful to provide more information on the computational setup, including the mesh size, and convergence criteria used. Additionally, you could elaborate on the boundary conditions applied in the simulations and how they were obtained.

·       Discuss the sensitivity analysis: It would be useful to discuss the sensitivity of the results to the input parameters used in the simulations, such as the excitation force and normal load. This would give readers a better understanding of the robustness of the results obtained.

·       Provide more details on the practical implications: While you briefly discuss the practical implications of your study, I suggest that you expand on this section. Specifically, you could discuss how the results obtained could be used to optimize the design of compressor stator blades or improve their maintenance.

·       Provide more context on the significance of the study: It would be helpful to provide more context on the significance of the study and why it is important to investigate the vibration response of compressor stator blades with contact friction.

·       provide more details on the assumptions made in the study, particularly with regards to the contact friction model used.

·       Provide some validation of your work. Compare the results to existing literature: It would be beneficial to compare the results obtained in your study to existing literature on the topic. This would allow readers to see how your study contributes to the current understanding of the vibration response of compressor stator blades.

·       There are some typos in the paper; correct them and reshape the paper for better understanding.

I recommended a Major revision 

Reviewer 2 Report

See attached comments.

The English is very poor. 

It must be proofread at the English Proofreading Institute.

Round 2

Reviewer 1 Report

Accepted

Reviewer 2 Report

None.